# Enhancing the therapeutic potential of P29 protein-targeted monoclonal antibodies in the management of alveolar echinococcosis through CDC-mediated mechanisms

Cuiying Zhang[1,2], Tao Li[3], Siyu Hou[1,2], Jing Tang[1,2], Rou Wen[1,2], Chan Wang[1,2], Shiqin Yuan[1,2,4], Zihua Li[1]*, Wei Zhao[1,2]*

1 School of Basic Medicine, Ningxia Medical University at Yinchuan, Yinchuan, Ningxia, China, 2 Ningxia Key Laboratory of Prevention and Control of Common Infectious Disease at Yinchuan, Yinchuan, China, 3 Department of Hepatobiliary Surgery, General Hospital of Ningxia Medical University at Yinchuan, Yinchuan, China, 4 Ningxia Eye Hospital, People's Hospital of Ningxia Hui Autonomous Region, Ningxia Medical University at Yinchuan, Yinchuan, China

* lizihua012178@126.com (ZL); zw-6915@163.com (WZ)

**Data Availability Statement:** All data are in the manuscript and Supporting information files.

## Abstract

Alveolar echinococcosis (AE) is a highly lethal helminth infection. Current chemotherapeutic strategies for AE primarily involve the use of benzimidazoles (BZs) such as mebendazole (MDZ) and albendazole (ABZ), which exhibit limited efficacy. In a previous study, the vaccine of recombinant *Echinococcus granulosus* P29 (r*Eg*P29) showed significant immunoprotection against *E. granulosus* in both mice and sheep. In the current study, we utilized hybridoma technology to generate five monoclonal antibodies (mAbs) against P29, among which 4G10F4 mAb exhibited the highest antigen-specific binding capacity. This mAb was selected for further investigation of anti-AE therapy, both in vivo and in vitro. In vitro, 4G10F4 inhibited a noteworthy inhibition of *E. multilocularis* protoscoleces and primary cells viability through complement-dependent cytotoxicity (CDC) mechanism. In vivo, two experiments were conducted. In the first experiment, mice were intraperitoneally injected with *Em* protoscoleces, and subsequently treated with 4G10F4 mAb (2.5/5/10 mg/kg) at 12 weeks postinfection once per week for 8 times via tail vein injection. Mice that were treated with 4G10F4 mAb only in dosage of 5mg/kg exhibited a significant lower mean parasite burden (0.89±0.97 g) compared to isotype mAb treated control mice (2.21±1.30 g). In the second experiment, mice were infected through hepatic portal vein and treated with 4G10F4 mAb (5mg/kg) at one week after surgery once per week for 8 times. The numbers of hepatic metacestode lesions of the 4G10F4 treatment group were significantly lower in comparison to the isotype control group. Pathological analysis revealed severe disruption of the inner structure of the metacestode in both experiments, particularly affecting the germinal and laminated layers, resulting in the transformation into infertile vesicles after treatment with 4G10F4. In addition, the safety of 4G10F4 for AE treatment was confirmed through assessment of mouse weight and evaluation of liver and kidney function. This study presents antigen-specific monoclonal antibody immunotherapy as a promising therapeutic approach against *E. multilocularis* induced AE.

**Funding:** This research was financially supported by funding from National Natural Science Foundation of China (82260331 to WZ, 82360402 to ZHL), Key Research and Development program of Ningxia Hui Autonomous Region (2022BSB03092 to ZHL), Science and Technology Support Project of Yinchuan (2023SF10 to TL) and Scientific Research Project of Ningxia Medical University (XZ2023027 to TL). The funder had no role in study design, data collection and analysis, decision to publish, or preparation of the manuscript.

**Competing interests:** The authors have declared that no competing interests exist.

## Author summary

Echinococcosis encompasses two significant zoonotic tapeworm diseases, cystic echinococcosis (CE) and alveolar echinococcosis (AE), caused by *Echinococcus granulosus* and *Echinococcus multilocularis*, respectively. AE is widely acknowledged as the most fatal helminth infection, with a mortality rate exceeding 90% within 10–15 years of diagnosis if left untreated or inadequately treated. Two benzimidazole carbamates, albendazole and mebendazole, are the only anti-infective drugs that are clinically efficient in interrupting the larval growth of Echinococcus spp. However, due to their toxicity and limited efficacy, there is a pressing need to explore novel therapeutic strategies for AE. In this study, a therapeutic antibody named 4G10F4 was produced against hydatid P29 using hybridoma technology. This candidate antibody significantly inhibited *E. multilocularis* both in vitro and in vivo, representing a potentially efficacious and safe antigen-specific monoclonal antibody immunotherapy for the treatment of AE. Additionally, it has been identified as a promising antibody drug and molecular target for the development of anti-AE therapy.

## Introduction

Echinococcosis encompasses two significant zoonotic tapeworm diseases, cystic echinococcosis (CE) and alveolar echinococcosis (AE), caused by *Echinococcus granulosus* and *Echinococcus multilocularis*, respectively [1]. CE is prevalent and widely distributed [2,3]. In regions where these diseases are endemic, the annual incidence of CE ranges from less than 1 to 200 cases per 100,000 individuals, while the incidence of AE ranges from 0.03 to 1.2 cases per 100,000 individuals [4]. AE is widely acknowledged as the most fatal helminth infection, with a mortality rate exceeding 90% within 10–15 years of diagnosis if left untreated or inadequately treated. After the diagnosis of AE, patients typically present with advanced disease characterized by symptoms and signs such as abdominal pain, jaundice, weight loss, and potential liver failure. Metacestode tissues exhibit uncontrolled proliferation through exogenous buds, leading to the invasion of neighboring tissues, thereby causing lesions to resemble tumors [5]. Although radical resection is the preferred treatment approach, complete excision of the tumor is often unattainable. Albendazole (ABZ) and mebendazole (MBZ) are the only available treatment options for patients who are unsuitable candidates for surgical intervention. Nevertheless, benzimidazoles exhibit parasitostatic properties, necessitating an extended treatment duration that may result in adverse hepatic effects, leukopenia, and hematuria [6]. Hence, it is imperative to develop novel therapeutic approaches to address these limitations and offer alternatives to conventional drug-based treatments for AE.

Monoclonal antibodies (mAbs), which are characterized as singular antibodies (Abs) derived from solitary B cells, have gained significant attention in the medical field. Currently, the US Food and Drug Administration (FDA) has approved over 100 mAbs for therapeutic intervention in numerous severe human diseases, thereby exhibiting the potential to revolutionize patient outcomes [7, 8]. Most mAbs that have received approval find applications within the domains of oncology and immunology. However, a limited number of mAbs have been developed specifically to combat infectious diseases, notably targeting the respiratory syncytial virus (RSV) with palivizumab, the anthrax toxin with Raxibacumab and Obiltoxaximab, and the bacterium *Clostridium difficile* with bezlotoxumab [9]. The lack of therapies utilizing mAbs to combat parasitic diseases is notable, with limited scientific literature documenting the use of therapeutic mAbs against parasitic pathogens. Within the realm of

parasitic diseases, two approaches can be used to develop and apply mAbs. One approach involves the utilization of antibodies that specifically target host antigens, particularly immune factors [10,11]. This approach enables the modulation of host immunity to enhance the efficacy of parasite elimination and mitigate the detrimental effects of hyperinflammation. An alternative approach involves the use of mAbs that directly target parasitic antigens, thereby inducing parasite elimination through diverse mechanisms, including antibody-dependent cellular cytotoxicity (ADCC), antibody-dependent cellular phagocytosis (ADCP), and complement-dependent cytotoxicity (CDC) [12]. The identification of appropriate and highly conserved targets for the development of therapeutic mAbs can be challenging and is of significant importance in the field of parasitic diseases, given the prevalence of antigenic variation in most parasites and the observed variability among different strains.

The P29 protein was first identified by Gonzalez as a novel 29 kDa antigen from *E. granulosus* while identifying parasite antigens distinct from those present in hydatid cyst fluid (HCF) [13]. Using a monoclonal antibody (mAb 47H. PS), Gonzalez demonstrated that P29, a protoscolex component, is primarily localized to the tegument and rostellum of protoscoleces as well as the germinal layer of the cyst. However, it was notably absent in the hydatid cyst fluids and adult worm extracts. This finding suggests a potential role for P29 as a developmentally regulated component of the *E. granulosus* metacestode. The P29 protein has been detected in the larval stages of both *E. granulosus* [14] and *E. multilocularis* [15]. Basic Local Alignment Search Tool (BLAST) analyses of *E. granulosus* and *E. multilocularis* P29 proteins revealed 99% identity (235/238 amino acids). Previous findings from our research indicate that recombinant *Eg*P29 (r*Eg*P29) protein elicits significant levels of specific IgG in vaccinated mice, with IgG1 and IgG3 being the predominant subclasses. Additionally, P29 protein demonstrates a strong protective immune response, with up to 96.6% efficacy against secondary *E. granulosus* infection in mice [16] and 94.5% immunoprotection in sheep, stimulating both Th1 and Th2 immune responses [17].

Despite extensive efforts to understand the precise functions of P29 in the parasite and its interactions with infected hosts, the absence of commercially accessible P29 monoclonal antibodies hinders our understanding of the molecular mechanisms governing P29 function and limits its potential clinical applications. In this study, we employed the traditional hybridoma technology to produce five mAbs against r*Eg*.P29 and examined their immunochemical properties. By evaluating the immunological affinity of r*Eg*.P29, we identified potentially valuable mAbs. Subsequently, we amplified, cloned, and characterized the mAbs. Furthermore, we assessed the antiparasitic efficacy of this mAb against *E.multilocularis* in vitro and in vivo.

## Results

### Generation of monoclonal antibodies against r*Eg*P29

The protein r*Eg*P29 was successfully expressed and purified as confirmed by SDS-PAGE, which showed a distinct band at approximately 30 kDa (S1 Fig). The generation of splenocyte-myeloma hybridomas was achieved using the standard hybridoma technique, followed by screening and selection based on their ability to produce anti-r*Eg*P29 monoclonal antibodies, as determined by enzyme-linked immunosobent assay (ELISA), and the flowchart was shown in Fig 1A. Purified r*Eg*P29 was utilized for the immunization of mice, resulting in the identification of the No.2 mouse's serum with the highest antibody titers against r*Eg*P29, reaching a dilution of 1:1,093,500 (S1 Table). Subsequently, spleen cells were collected from mouse No. 2 and fused with Sp2/0 myeloma cells using electrofusion. Following screening and subcloning, five hybridomas secreting monoclonal antibodies against *Eg*P29 were successfully identified and designated as 2G9G2, 1F9F10, 1F9B12, 4G10F4, and 1F9B3 (S2, S3 and S4 Tables).

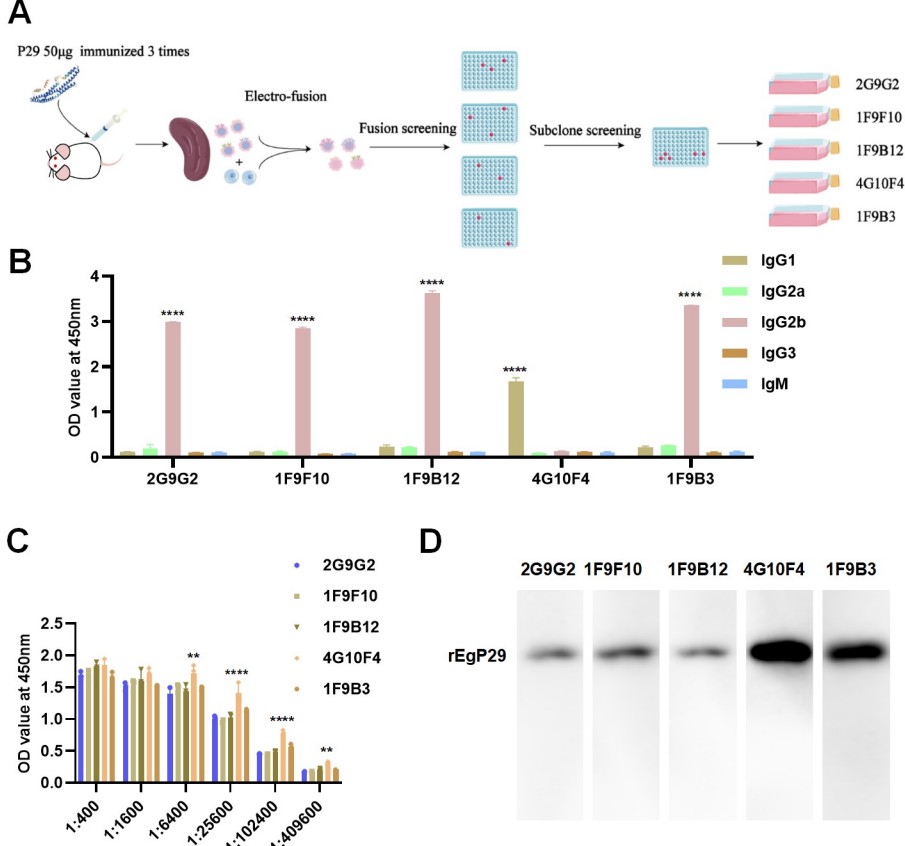

**Fig 1. Generation of monoclonal antibodies against r*Eg*P29.** (A) Flow chart for the preparation of anti-r*Eg*P29 positive hybrydoma cell lines. (B) Subtypes of each mAb were determined by ELISA. (C) The antibody titers of each mAb were measured by ELISA. (D) Each mAb recognized the r*Eg*P29 protein. **p < 0.01, ****p < 0.0001.

## Subtype and recognition of the monoclonal antibodies

The results obtained from the indirect ELISA analysis indicated that mAbs 1F9B3, 1F9B12, 1F9F10, and 2G9G2 belonged to the IgG2b subtype, while 4G10F4 belonged to the IgG1 subtype (Fig 1B). The protein concentration of the culture supernatant from the five hybridoma cell lines was quantified utilizing the BCA method. Subsequently, the protein levels were normalized for titer analysis and western blot analysis. Furthermore, the titers of all five mAbs exceeded 1:102,400 (Fig 1C) and specific recognition of the target antigen r*Eg*P29 was confirmed (Fig 1D). Notably, mAb 4G0F4 exhibited the highest titer, and was selected for subsequent experiments.

## Specificity of 4G10F4 mAb

The 4G10F4 hybridomas were cultured on a large scale, followed by harvesting and purification of the mAbs, as depicted in Fig 2A. Next, we examined the specific binding of 4G10F4 mAb to *Eg* materials. The mAb binded to the protoscoleces (PSCs), and hydatid cyst wall (HCW), but not hydatid cyst fluid (HCF) (Fig 2B). Subsequently, the localization of P29 in PSCs and HCW was determined by indirect immunofluorescence staining with this mAb, as illustrated in Fig 2C. Red fluorescence was observed in the germinal layer of the hydatid cyst membrane, as well as in the tegument, rostellum, and suckers of the protoscoleces.

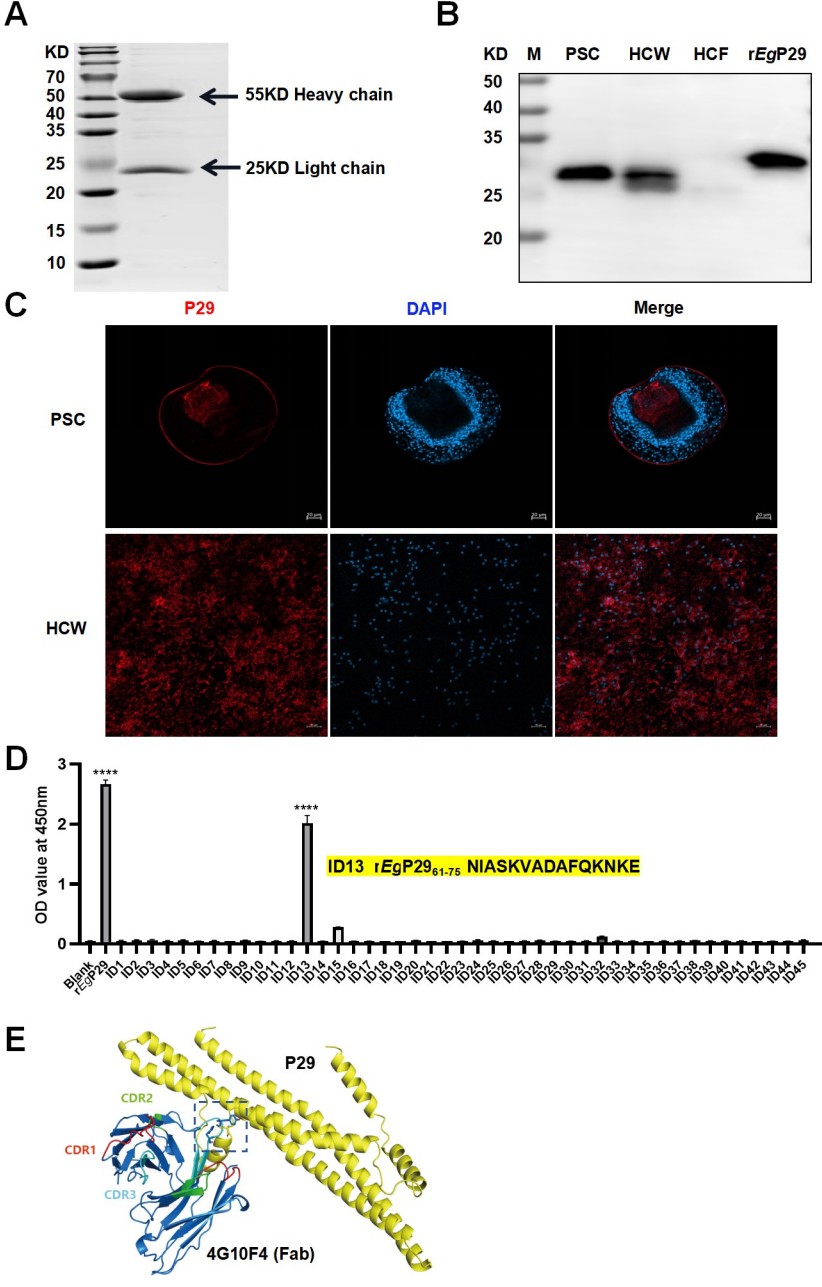

**Fig 2. Purification and characterization of 4G10F4 mAb.** (A) SDS-PAGE analysis of purified 4G10F4 mAb from hybridoma cell culture supernatant. (B) 4G10F4 mAb reacted with proteins from *E. granulosus* materials. M: protein marker, Lane PSC: protosoleses; Lane HCW: hydatid cyst wall; HCF: hydatid cyst fluid; Lane r*Eg* P29: recombinant *echinococcus granulosis* P29. (C) Confocal microscopy images of the hydatid cyst P29 recognized by 4G10F4 mAb. (D) Epitope identification of 4G10F4 mAb by peptide scanning. The core sequence is [61]NIASKVADAFQKNKE[75]. (E) The putative three-dimensional structures of *Eg* P29 and the anti-P29 antibody were generated, and the binding site between the antigen and antibody was predicted. The black dotted line framed area indicates location of antigen-antibody interaction site.. ****$p < 0.0001$.

**Table 1. Nucleotide and amino acid sequences of complementary determining regions (CDRs) of 4G10F4 mAb.**

| Chain Type | Sequence | CDR1 | CDR2 | CDR3 |
|---|---|---|---|---|
| VH | Nucleotidesequence | GGGTTTTCATTATTCAGCTATGCT | ATATGGGCTGGTGGAAGCACA | GCCAGTATTCATTACTACGGCTA CTGGCACTTCGATGTC |
| | Translation | GFSLFSYA | IWAGGST | ASIHYYGYWHFDV |
| VL | Nucleotidesequence | cagagcattgtacatagtaatggaaacacctat | aaagtttcc | tttcaaggttcacatgttccgtacacg |
| | Translation | QSIVHSNGNTY | KVS | FQGSHVPYT |

## Characteristics of 4G10F4 mAb: Epitope identification, sequence analysis and affinity determination

Among the 45 synthesized overlapping peptides, peptide 13 exhibited strong reactivity with 4G10F4 mAb, and the core sequence of the peptide was identified as [61]NIASKVA-DAFQKNKE[75] (Fig 2D). The sequencing results of the heavy chain (H) and light chain (L) of the 4G10F4 mAb can be found in additional files (S1 and S2 Files). Subsequently, the IgBLAST tool (http://www.ncbi.nlm.nih.gov/igblast/) was used to analyze the sequences, resulting in the prediction of nucleotide and amino acid sequences corresponding to complementary demining regions (CDRs) of the H and L domains, as shown in Table 1. Based on the amino acid sequences of 4G10F4 and P29 (GenBank: AAD53328.1), the rigid protein-protein docking analysis between 4G10F4 mAb and P29 was conducted utilizing gramm-X to investigate their interrelationship, as shown in Fig 2E. The protein structure was sourced from the Uniprot (www.uniprot.org), PDB (RCSB PDB: Homepage), and Alphafold databases. Pymol (Version 2.4) was employed for the examination of protein interactions and subsequent visual analysis. CDRs were labeled on the antibody structure and closed to the screened epitope on P29. Upon closer examination of protein-protein interactions (S2 Fig), hydrogen bonds were observed between the antibody and P29, facilitated by amino acid residues (indicated by yellow dashed lines) such as TYR-103, HIS-105, and ASN-75, resulting in a binding energy of -8.1 kcal/mol. This analysis indicated the formation of a stable protein docking model between the antibody and P29. Specifically, TYR-103 and HIS-105 amino acid residues were found within the CDRs region of the mAb, while the ASN-75 residue was situated within the identified linear epitope of 61–75 amino acid sequence. Surface plasmon resonance (SPR) analysis was conducted to determine the affinity constants (KD) of 4G10F4 mAb towards r*Eg*P29. The results revealed that 4G10F4 exhibited potent binding to r*Eg*P29, with a KD value of 6.29nM (Table 2 and S3 Fig).

## Effects of 4G10F4 on *E.multilocularis* protoscoleces in vitro

In order to assess the viability of protoscoleces, we employed eosin and trypan blue staining techniques, alongside CCK8 and LDH cytotoxicity detection methods. Eosin and trypan blue staining are commonly utilized approaches for quantifying the viability proportion of protoscoleces, by determining the ratio of deceased to living protoscoleces [18]. The CCK8 assay quantifies dehydrogenase activity in living cells, thereby determining the number of viable cells. Conversely, the LDH assay kit measures the release of LDH from damaged cell

**Table 2. Affitiny of P29 antibody to r*Eg* P29 measured by SPR.**

| Antigen | Ligand | ka(1/Ms) | kd(1/s) | $KD$(M) |
|---|---|---|---|---|
| r*Eg* P29 | 4G10F4 | $7.78 \times 10^4$ | $4.9 \times 10^{-4}$ | $6.29 \times 10^{-9}$ |

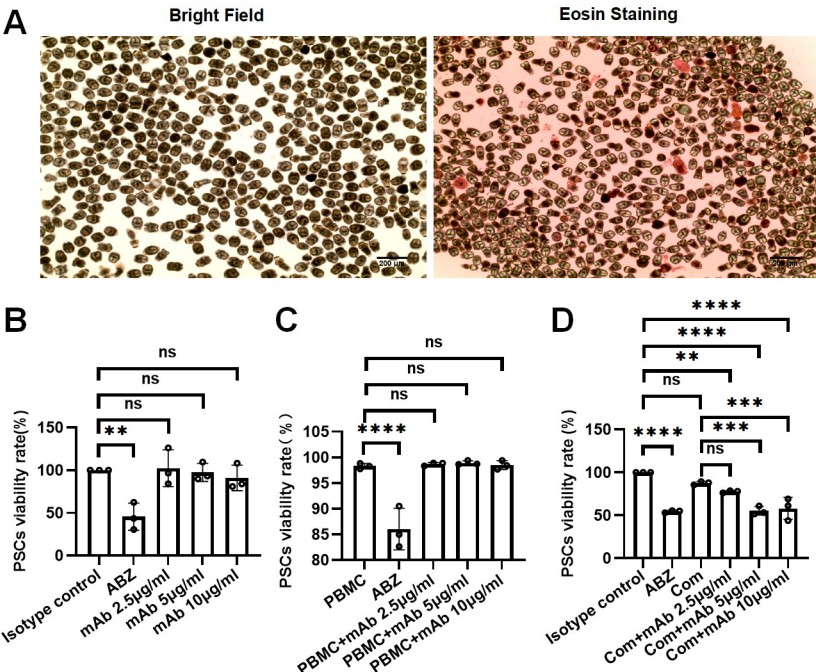

**Fig 3. The in vitro effects of 4G10F4 on the viability of E. multilocularis protoscoleces.** (A) The viability of protoscoleces were determined by eosin exclusion. (B) Direct effects of 4G10F4 on the viability of protoscoleces. (C) Effects of 4G10F4 on the viability of protoscoleces in the present of PBMC. PBMC was used as negative control. (D) Effects of 4G10F4 on the viability of protoscoleces in the present of complement. ABZ was used as positive control in protoscoleces viability analysis. ABZ: albendazole; PBMC: peripheral blood mononuclear cell; Com: complement.
**p < 0.01, ***p < 0.001, ****p < 0.0001, ns: no statistical significance.

membranes, confirming the number of dead cells and assessing cytotoxicity. Prior to the experiment, the viability of *E. multilocularis* PSCs was assessed by eosin exclusion. The results indicated that the viability of PSCs was greater than 95% (Fig 3A). Fig 3B demonstrates that ABZ, as a positive control, significantly inhibited the viability of PSCs in vitro, while three concentrations (2.5 µg/ml, 5 µg/ml, 10 µg/ml) of the anti-P29 monoclonal antibody 4G10F4 did not directly induce any killing effect on PSCs (evaluated by CCK8 assay). ADCC activity induced by 4G10F4 was evaluated using murine peripheral blood mononuclear cells (PBMCs) as effector cells. As shown in Fig 3C (evaluated by trypan blue staining) and S4D Fig (evaluated by LDH-based cytotoxicity detection), 4G10F4 did not elicit ADCC activity against *E. multilocularis* PSCs. However, the addition of a complement to the system enhanced the cytotoxic effect of the antibody (Fig 3D), as evaluated using the CCK8 assay. Specifically, 4G10F4 at concentrations of 5 µg/ml and 10 µg/ml obviously inhibited PSCs viability. Visually, there was no detectable difference between the PSCs exposed to different treatments(S4A, S4B and S4C Fig). These results suggest that the killing effects of 4G10F4 were partially dependent on complement-dependent mechanisms.

## Effects of 4G10F4 on the viability of *E. multilocularis* primary cells in vitro

To further investigate the effects of 4G10F4 on CDC, we isolated primary *E. multilocularis* cells from in vitro cultivated secondary vesicles derived from *E.multilocularis* metacestodes and performed in vitro cell-killing assays. As shown in Fig 4A, the control and antibody-treated groups exhibited discernible cloned cell clusters formed by *E. multilocularis* primary

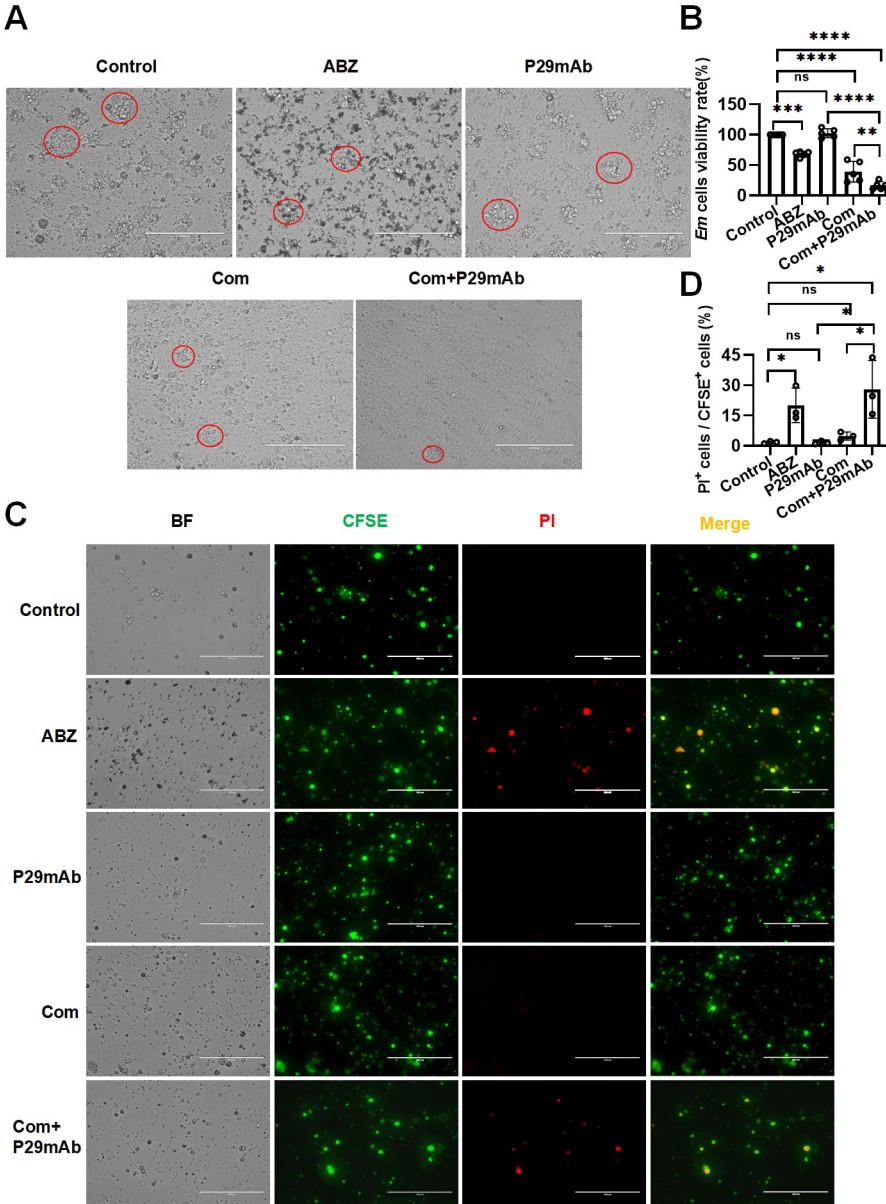

**Fig 4. 4G10F4 showed CDC activity on the E. multilocularis primary cells.** (A) Cell growth was visualised under a light microscope after 24 hours of intervention. (B) Cell viability was assessed via CCK8 assay. (C) Live cells stained with CFSE (green) and dead cells stained with PI (red) were observed using fluorescence microscope. (D) The percentage of the dead cells were calculated using Image J software. Red circled areas indicate cell clones. *Em*: *Echinococcus multilocularis*; ABZ: albendazole; Com: complement. *p < 0.05, **p < 0.01, ***p < 0.001, ****p < 0.0001, ns: no statistical significance.

cells, whereas the ABZ- and complement-treated groups displayed a reduction in the size of these cell clusters. Notably, the anti-P29 mAb plus complement-treated group showed a near absence of clones formed by *E. multilocularis* primary cells. Assessment of cell viability using the CCK8 assay revealed that treatment with the anti-P29 mAb did not affect cell viability. However, in the group treated with anti-P29 mAb in combination with complement, there was a substantial decrease in cell viability, which was significantly lower than that observed in

the group treated with complement alone (Fig 4B). The percentage of cell death after treatment was determined by 5-(6)-carboxy-fluorescein succinimidyl ester (CFSE) and propidium iodide (PI) double staining. CFSE-labeled *E. multilocularis* cells emitted green fluorescence, whereas dead cells stained with PI emitted red fluorescence, as depicted in Fig 4C. The ABZ and Complement+P29 mAb groups exhibited significant increases in cytotoxicity compared with the control group, whereas the other two intervention groups did not. The proportion of dead cells of the Complement+P29 mAb group (28% ± 14.31%) was significantly higher than that of the P29 mAb group (1.53% ± 0.78%) and the Complement group (4.82% ± 2.02%) (Fig 4D). These results indicate that 4G10F4 has a cytotoxic effect on *E. multilocularis* cells in the presence of the complement system.

## 4G10F4 reduced parasite burdens in intraperitoneal infected mice with *E. multilocularis*

To assess the effects of 4G10F4 on *E. multilocularis* in vivo, we treated *E. multilocularis* intraperitoneally infected mice with 4G10F4 mAb. At 12 weeks post-infection, when metacestode lesions reached 3-5mm$^3$ in the abdomen through B ultrasound detection in the model mice (S5 Fig), treatment was initiated. As shown in Fig 5A, 4G10F4 mAb was administered at doses of 2.5, 5, 10 mg/kg via tail vein injection once a week for 8 weeks. At the end of the experiment, the mice were euthanized and the metacestode lesions were removed and weighed, after which metacestode inhibition rates were calculated. The metacestode lesion images are shown in Fig 5B. No visible lesions were found in two out of nine mice in the mAb (2.5 and 5 mg/kg) and ABZ treatment groups. Metacestode lesion weights were compared between groups (Fig 5C). The mean weight of the metacestode was significantly lower in the mAb 5 mg/kg group (0.89 ±0.97 g) and the ABZ group (0.81±1.27 g) than that in the isotype control group (2.21±1.30 g). A slight decrease in wet weights of the metacestode was seen in mAb 2.5 mg/kg and mAb 10 mg/kg groups(1.96±1.68 g and 1.56±1.71 g), but did not reach statistical significance compared with the isotype control. During the whole treatment, we monitored no abnormal change in the body weight of animals treated (S6E Fig). The spleen and liver indices were calculated as described in materials and methods section. The spleen indices of the four intervention groups were significantly lower than that of the control group(Fig 5D), whereas the liver index decreased only in the ABZ treatment group (Fig 5E). Based on the wet weights of the metacestode in each group (Fig 5C), metacestode inhibition rates were calculated as described in materials and methods section. In the present study, 4G10F4 exhibited metacestode inhibition rates of 11.11%, 59.70%, and 29.31% at 2.5, 5, and 10 mg/kg, respectively. ABZ showed a metacestode inhibition rate of 63.24% at 50 mg/kg, suggesting that the anti-parasite effect of 4G10F4 was comparable to that of ABZ.

## 4G10F4 did not adversely affect hepatic or renal functions

To investigate the effect of 4G10F4 on hepatic and renal functions in mice, serum levels of alanine aminotransferase (ALT), aspartate aminotransferase (AST), creatinine (CREA), and blood urea nitrogen (BUN) were assessed. Notably, the activities of ALT and AST were markedly elevated in the isotype group compared to those in the blank group, whereas all intervention groups exhibited a decrease in these activities, approaching normal levels(S6A and S6B Fig). However, there were no significant differences in the serum levels of CREA or BUN among the various groups(S6C and S6D Fig). These data suggested that 4G10F4 did not adversely affect hepatic or renal function in the experimental mice.

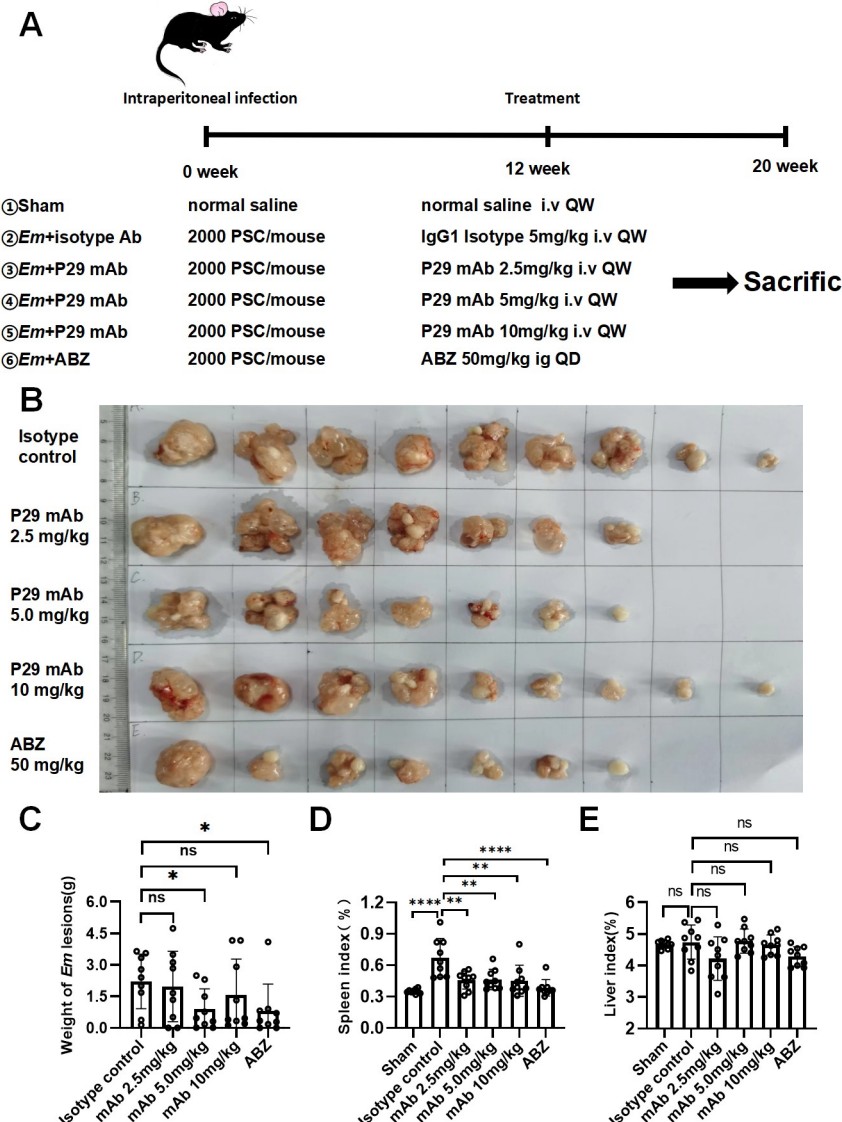

**Fig 5. Evaluation of therapy effects of 4G10F4 on *E.multilocularis* intraperitoneal infected mice.** (A) Schematic presentation of animal experimental protocol for the 4G10F4 anti-parasitic effects on Em intraperitoneal infected mice (n = 9/group). (B) Gross morphology of metacestode lesions. (C) Quantitative weight analysis of the E.multilocularis metacestode tissues. (D and E) Comparison of spleen and liver index between groups after treatment. *Em*: *Echinococcus multilocularis*; PSC: protoscolex; ABZ: albendazole; i.v: intravenous; ig: intragastric gavage; QW: once a week; QD: once a day. *p < 0.05, **p < 0.01, ***p < 0.001, ****p < 0.0001, ns: no statistical significance.

## 4G10F4 reduced the inflammation cytokines to normal levels

To analyze the immune effects induced by 4G10F, we used CBA mouse inflammation kit to examine the levels of the cytokines: Interleukin-6 (IL-6), Interleukin-10 (IL-10), Monocyte Chemoattractant Protein-1 (MCP-1), Inerferon-γ(IFN-γ), Tumor Necrosis Factor (TNF), and Interleukin-12p70 (IL-12p70) in serum(S7A–S7F Fig). Compared with the blank group, IL-6, IL-10, MCP-1, IFN-γ, TNF and IL-12p70 levels of the isotype control group elevated significantly. However, treatment with 4G10F4 at the three doses decreased the levels of the six inflammatory cytokines. ABZ markedly decreased the IL-10, IFN-γ, TNF and IL-12p70 levels.

## 4G10F4 reduced parasite burdens in infected mice through portal vein with *E. multilocularis* protoscoleces

The efficacy of 4G10F4 against *E. multilocularis* was evaluated in mice that were intraperitoneally infected with *E. multilocularis* protoscoleces. To confirm the in vivo antiparasitic effectiveness of this monoclonal antibody, we used a mouse model of liver infection via the hepatic portal vein with *E. multilocularis* protoscoleces, which closely mimics the natural pathway of parasite transmission. The experimental protocol depicted in Fig 6 involved the division of ten surviving mice post-surgery into two groups: a control group treated with isotype IgG1 and a group treated with 4G10F4 monoclonal antibody (5mg/kg). Treatment was administered one week post-infection and continued weekly for a duration of 8 weeks. Subsequently, at the end of the treatment, B-mode ultrasound examinations were conducted on all experimental mice, resulting in the visualization and presentation of liver lesions (Fig 7A). A comparison between the isotype control group and the 4G10F4 treatment group revealed a reduction in both the quantity and size of the hepatic lesions. At the eight-week mark following treatment, the mice were euthanized, and a visual representation of the macroscopic lesions observed in the liver is depicted in Fig 7B. The number of lesions present on the surface of the liver was significantly lower than that in the isotype control mice that received a sham treatment with an unrelated antibody of the same isotype (Fig 7C). Pathological alterations in the liver were identified using hematoxylin and eosin (H&E) and Periodic Acid-Schiff (PAS) staining (Fig 7F). Periodic acid-Schiff (PAS) staining of the mucopolysaccharides was performed to visualize the laminated layers inside the parasitic vesicles. In the control group, approximately half of the liver was replaced by metacestode tissue, and the pseudo-tumor parasitic mass contained numerous vesicular structures embedded within the dense fibrous tissue. The hepatic parenchyma adjacent to the lesions experienced progressive invasion by the fibrous connective tissue septa. Parasitic vesicles retained portions of the laminated and germinal layers along with a considerable number of protoscoleces. In the 4G10F4 treatment group, the lesions experienced a notable reduction and there was a noticeable deposition of collagen in the host tissues surrounding the metacestodes. PAS staining revealed a scarcity of laminated layers, and protoscoleces were barely discernible within the parasitic vesicles.

The liver index increased in the isotype control group and decreased in the mAb-treated group, with no statistically significant differences observed between these groups (Fig 7D). Conversely, the spleen index demonstrated a significant decrease in the mAb-treated group compared to that in the isotype control group (Fig 7E). Throughout the treatment period, no adverse reactions were observed in the mice, and there were no abnormal fluctuations in the body weights of the treated animals (S6F Fig).

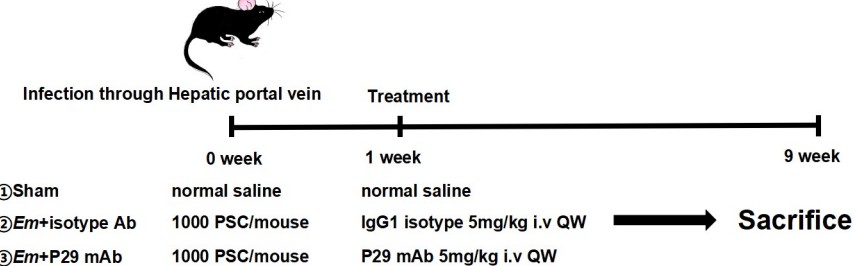

**Fig 6. Schematic presentation of animal experimental protocol for the 4G10F4 anti-parasitic effects on infected mice through portal vein (n = 5/group).** *Em*: *Echinococcus multilocularis*; PSC: protoscolex; i.v: intravenous; ig: intragastric gavage; QW: once a week; QD: once a day.

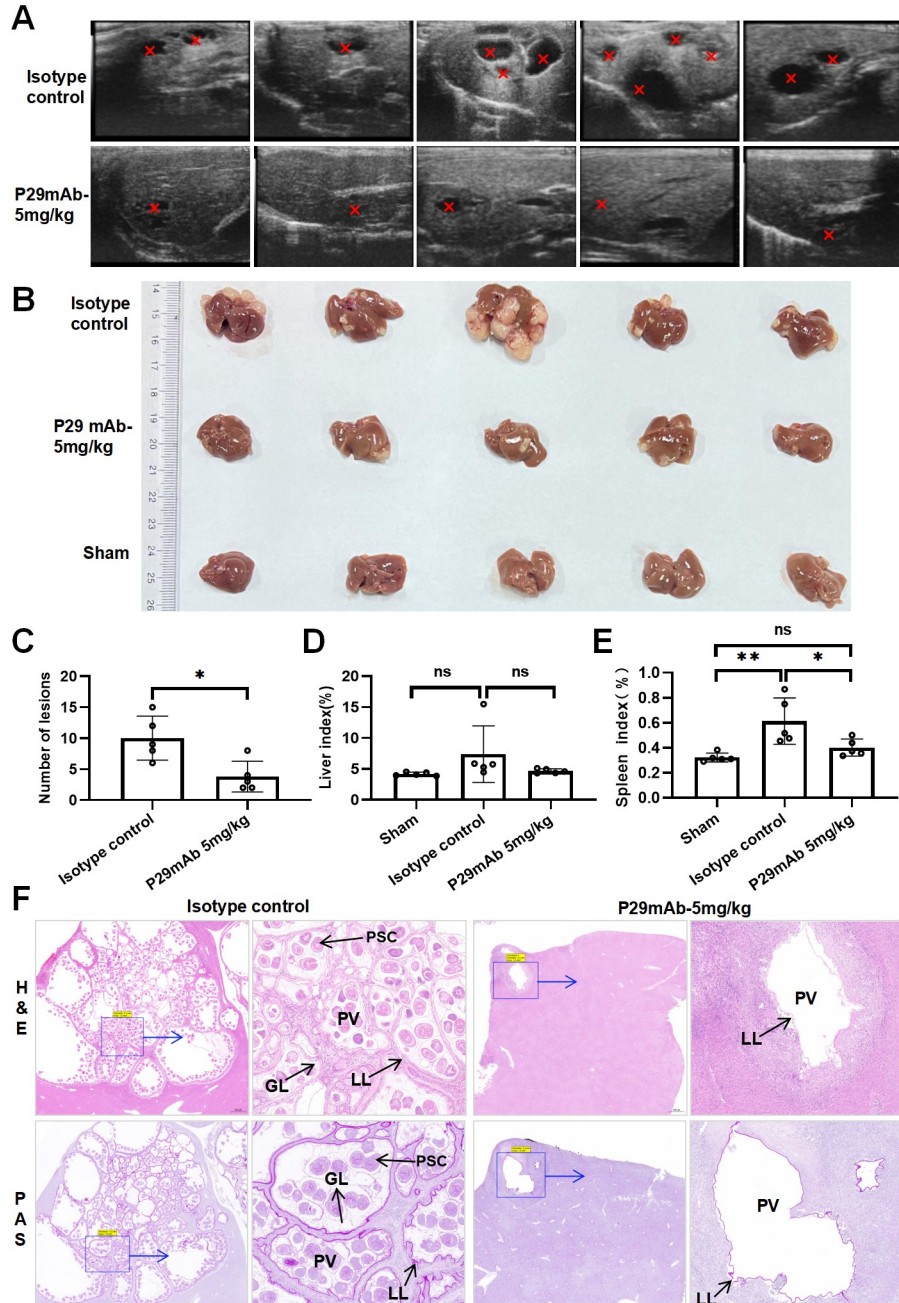

**Fig 7. Evaluation of therapy effects of 4G10F4 on *E. multilocularis* infected mice through portal vein.** (A) Murine liver B-ultrasound images in different groups. The red arrows indicate the metacestode lesions. (B) Gross morphology of metacestode lesions in liver. (C) Comparison of the numbers of infectious foci in the liver surface between groups after treatment. (D and E) Comparison of spleen and liver index between groups after treatment. (F) Histopathological changes were observe by H&E staining and PAS staining. H&E: hematoxyli and eosin; PAS: Periodic Acid-Schiff; GL: germinal layer; LL: laminated layer; PSC: protoscolex; PV: parasitic vesicle. *p < 0.05, **p < 0.05, ns: no statistical significance.

## Discussion

Previous studies have indicated that antibiotics (mefloquine and amphotericin B) [19, 20], Chinese herbal medicines (crocin) [21], protease inhibitors (bortezomib and imatinib) [22,23], and immune checkpoint inhibitors (anti-PD-1/PD-L1 drugs and anti-TIGIT drugs) [11,24,25] exhibit anti-echinococcosis effects in animal models. However, therapeutic antibodies targeting specific AE markers have not been identified. Our previous study demonstrated that r*Eg*P29 exhibits a favorable immunoprotective effect in both sheep and mice [16,17]. Additionally, our findings revealed a notable increase in serum IgG levels against r*Eg*P29 among patients with CE2 type cysts compared to healthy donors [26]. These findings indicate a potential correlation between the humoral immune response triggered by P29 and protection against echinococcosis.

In this study, we employed traditional hybridoma technology to generate five mAbs targeting P29. Among the five generated mAbs, four belonged to the IgG2b subtype, whereas 4G10F4 belonged to the IgG1 subtype. Humans and mice both produce four distinct subclasses of IgG heavy chains, known as IgG1-4 in humans and IgG1, 2a (or 2c in certain inbred strains), 2b, and 3 in mice [27]. The binding of these antibodies to class-specific Fc receptors (FcRs) on leukocytes is a crucial mechanism through which antibodies trigger an adaptive immune response to eliminate infected or malignant cells. The interactions between the Fc region of IgG and FcγR determine the binding affinity of IgG subclasses to FcγRs, thereby conferring unique biological activities. Specifically, IgG1 exhibits higher affinity for FcγRIIa but lower affinity for FcγRIIIa and FcγRIIIb compared to IgG3 [28]. Therefore, IgG1 demonstrates the ability to facilitate clinically relevant processes such as antibody-dependent cell-mediated cytotoxicity (ADCC) and antibody-dependent cellular phagocytosis (ADCP). Additionally, IgG1 forms hexamers on the surface of target cells, enabling it to activate complement and induce complement-dependent cytotoxicity (CDC). In contrast, IgG2 exhibits diminished effector functions compared to IgG1 and IgG3 due to its lower affinity for Fcγ receptors. Complement activation by IgG2 is contingent upon high antigen density on the target cell [28]. Consequently, IgG2 is considered a less immunologically active isotype that may be beneficial for therapeutic interventions requiring precise blockade mechanisms.

The 4G10F4 antibody with the highest titer was selected for further investigation. Despite the utilization of prokaryotic-expressed P29 protein as the immunogen, 4G10F4 exhibited recognition not only towards recombinant P29 but also towards the natural P29 protein derived from the parasite. Our findings provide evidence that P29 is present in the germinal layer of the hydatid cyst membrane, as well as in the tegumentary syncytium, rostellum, and protoscoleces. However, P29 was absent in the hydatid cyst fluid, which aligns with the findings of Gonzalez [13]. These results suggest that P29 is a component of the metacestode of *E. granulosus* which is regulated during development and is not secreted in vivo. Its localization in the protoscolex region, which is essential for interaction with the host mucosal epithelium, implies its potential involvement in nutrient absorption, evasion of immune surveillance, and invasion of host tissue. However, further research is required to fully understand the exact function of P29. It is imperative to acknowledge that an mAb clone recognizes a singular domain rather than the entire molecule. This particular domain, known as the B cell epitope, is specifically recognized by B cell antigen receptors or antibodies and serves as the fundamental structure for eliciting a targeted humoral immune response. Identification of antigenic epitopes plays a crucial role in understanding the intricate relationship between antibodies and antigens in terms of structure and function. Moreover, it offers valuable insights into antibody engineering and design of therapeutic antibodies. In this study, peptide scanning was conducted using 45 overlapping peptides of r*Eg*P29, revealing that the 4G10F4 mAb recognized the core

sequence [61]NIASKVADAFQKNKE[75]. Swiss model structure analysis indicated that this epitope was situated within a loop region of P29, which aligned closely with the predicted binding site for the antigen-antibody interaction. It is commonly postulated that loop regions serve as binding sites for functional antibodies and exhibit notable specificity and accessibility, as supported by previous studies [29,30].

Hybridomas exhibit the capacity to provide a continuous supply of mAbs with high affinity and specificity. However, the maintenance of hybridoma cells is associated with antibody gene loss, insufficient cell viability, reduced antibody production, and potential cell death. To overcome these limitations, we cloned and sequenced H and L chain genes derived from the 4G10F4 hybridoma and subsequently predicted the amino acid sequences within their complementarity-determining regions (CDRs). This notable undertaking signifies a crucial advancement in the field of engineering P29 monoclonal antibodies and their derivatives, thereby enhancing their applicability in the domains of prevention, diagnosis, and treatment, as well as in the investigation of the interaction between the parasite and its host.

The P29 protein/antigen exhibited a significant level of conservation across various species and genotypes within the *Echinococcus* genus [31]. In this study, we aimed to evaluate the therapeutic efficacy of an anti-r*Eg*P29 mAb for the treatment of AE. Encouragingly, we observed distinct anti-parasitic effects of this mAb both in vivo and in vitro. It is well established that antibodies can combat pathogenic infections through various mechanisms. These mechanisms include the direct action of antibodies, primarily mediated by fragmented antigen-binding domains (Fabs), that involve receptor blockade or neutralization. Additionally, immune-mediated cell killing mechanisms, such as complement-dependent cytotoxicity (CDC) and antibody-dependent cellular cytotoxicity (ADCC), rely on the fragment crystallizable (Fc) domain. In our in vitro experiments, we initially evaluated the direct anti-parasite effects of mAbs on *E. multilocularis* protoscoleces at three different mAb concentrations. No effects were observed on the viability of protoscoleces in our cultivation system at any concentration of treatment. In light of the intricate mechanisms involved in antibody-mediated killing, we introduced PBMC and complemented the system to assess the therapeutic efficacy of the anti-P29 mAb in vitro. Our findings demonstrated that concentrations of 5μg/ml and 10μg/ml of the mAb exhibited remarkable killing effects on PSCs in the complement system, whereas the three concentrations of the mAb had no impact on the viability of PSCs in the PBMC system.

The cellular mechanism by which anti-P29 mAb acts against *E. multilocularis* was confirmed in this study. We employed an improved method to isolate and cultivate primary cells derived from *E. multilocularis* tissues and verified the absence of host cells in the isolated cell population using 18S rRNA gene sequencing, as described in previous study [32]. The germinal layer of *E. multilocularis* metacestode encompasses diverse cell types, such as undifferentiated germinative cells, as well as differentiated cell types, including tegumental cells, muscle cells, glycogen/lipid-storing cells, and nerve cells [33–35]. Koziol [35] characterized germinative cells in metacestode and primary cell cultures as the only proliferating cells driving metacestode growth and regeneration. In this experiment, it was observed that the 4G10F4 mAb did not directly affect cell viability when administered alone. However, in the presence of complement, the mAbs significantly reduced the viability of *E. multilocularis* cells. Furthermore, this treatment induces cell death and hinders clone formation. These findings confirmed that 4G10F4 mAb possesses parasiticidal properties in vitro, with CDC serving as a crucial mechanism for its lethal effects.

Compounds that exhibit promising in vitro efficacy can be further assessed using in vivo models. In this study, two infection models were employed: (i) intraperitoneal infection of mice with *E. mutilocularis* metacestode PSCs, and (ii) infection of mice with PSCs via the hepatic portal vein. For the secondary infection model, the determination of parasite weight or

lesion number and size has been demonstrated to be as reliable as non-invasive imaging techniques, providing a definitive measure of in vivo drug effectiveness against AE [36]. The inhibitory effects of 4G10F4 on the growth of *E. multilocularis* metacestodes were evident in both the experimentally infected mouse models. The intraperitoneal injection method for protoscolices demonstrated a high survival rate of mice, however, the success rate of infection varied. As a result, treatment initiation at 12 weeks postinfection, when metacestode lesions were visible, was deemed appropriate. The administration of 4G10F4 mAb for established infection showed promising potential for clinical application. Prior research has shown that C57/BL mice infected with high doses (HD, 1000, 2000) of live PSCs via the portal vein have a nearly 100% success rate in developing metacestode lesions in the liver [37]. Therefore, mAb treatment was administered to these hepatic infected mice one week post-infection, during the acute infectious stage. The use of infection through the portal vein closely resembles the natural infection process, thus enhancing the reliability of the experimental outcomes. Following treatment with the 4G10F4 monoclonal antibody, the volume and number of metacestode lesions were significantly reduced, accompanied by distinct changes in their internal structures. In mice with hepatic infections, the lesions consisted of multiple separate cysts, with the innermost germinal layer of the cyst producing brood capsules and protoscoleces in the central cavity. In contrast, the administration of 4G10F4 resulted in a reduction in hepatic lesions, accompanied by a significant influx of inflammatory cells and an increase in fibrous connective tissue. These alterations have the potential to impede the proliferation of *E. multilocularis* within the host through endogenous and exogenous budding reproduction around them [21]. Furthermore, inflammatory cell infiltration was accompanied by significant disruption of the germinal layer, leading to the formation of infertile parasitic vesicles. These findings strongly indicate the potential parasiticidal efficacy of the 4G10F4 monoclonal antibody against alveolar echinococcosis in an in vivo.

Cytokines and chemokines play pivotal roles in severe and life-threatening illnesses by connecting innate and adaptive immunity and serving as crucial regulatory factors in inflammatory diseases [38]. Throughout the duration of AE, individuals with cured, stable, or progressive disease exhibit distinct response profiles to regulatory and inflammatory cytokines and chemokines [39]. However, there is limited knowledge regarding the factors that may impede the survival and proliferation of metacestodes or eradicate *E. multilocularis* infections. A contrasting cytokine secretion profile was observed in patients with abortive versus progressive AE, with a prevalence of interleukin (IL)-10, IL-5, and interferon (IFN)-γ associated with progressive AE [40]. Conversely, murine experimental studies have indicated that resistance to *E.multilocularis* growth is associated with significant and sustained cellular proliferation and elevated levels of Th1 type cytokines [41]. The results of our experiment demonstrated that the levels of IL-6, IL-10, MCP-1, IFN-γ, TNF, and IL-12p70 were significantly increased in mice infected with *E. multilocularis* compared to non-infected control mice. Treatment with 4G10F4 and ABZ effectively decreased the levels of the six inflammatory markers compared with those observed in non-infected control mice. Despite the notable inter-individual variations in the concentrations of these six inflammatory cytokines, our comparative analysis revealed no correlation between this variation and the size of the parasitic lesions in each mouse. Moreover, the effective treatment group exhibited recovery of spleen and liver indices, indicating a reduction in inflammation and improvement in *E. multilocularis* infection with the administration of anti-P29 mAb. This phenomenon may have implications for the anti-inflammatory properties induced by the mAb targeting P29. However, the specific roles of P29 remain unclear, further investigations are required to elucidate the exact underlying mechanisms. Additionally, the toxicity of 4G10F4 in AE treatment was assessed by monitoring mouse weight and evaluating liver and kidney function, confirming its safety.

The implementation of 4G10F4 as a therapeutic antibody in the anti-AE field is still in the early stages of development. Many unanswered questions remain, such as the selection of an effective dosage (e.g., why a high dose of 10 mg/kg did not produce the desired effect with 4G10F4) and the need for further investigation into the exact mechanism of action. In addition, numerous other issues necessitate further research. In our study, we observed a metacestode growth inhibition of 59.70% with 4G10F4, indicating its potential for further improvement. Combination therapy has emerged as a significant approach in the realm of cancer therapy with the aim of enhancing response rates and combating resistance. The efficacy of anti-P29 monoclonal antibody (mAb) in conjunction with ABZ has been assessed in murine infection models. In addition to IgG, antibody drug conjugates (ADCs) represent prevalent modalities for anticancer antibodies [42]. In future investigations, it may be worthwhile to explore the conjugation of 4G10F4 with an antiparasitic medication, such as ABZ, to augment therapeutic efficacy. Furthermore, the generation of monoclonal antibodies targeting *E. multicularis* P29 is underway to evaluate their anti-parasitic effects and to compare them with those of 4G10F4. Additionally, ongoing investigations are exploring potential modifications of anti-P29 mAbs to enhance their therapeutic efficacy.

In conclusion, we effectively generated five novel mAbs targeting r*Eg*P29 that exhibit distinct recognition activities. Notably, 4G10F4, an IgG1 mAb, demonstrated the most potent antigen-specific binding capacity and was subsequently confirmed to possess therapeutic efficacy against *E. multilocularis* both in vitro and in vivo. It is plausible that the complement-dependent cytotoxicity (CDC) pathway contributes to the anti-parasitic action of 4G10F4; however, further investigation is warranted to elucidate the precise underlying mechanisms.

## Materials and methods

### Ethics statement—Animals

All animal studies conducted in this research were granted approval by the Experimental Animal Ethics Committee of Ningxia Medical University (2022-G139). Female BALB/C mice with an average weight of 18±2g were procured from the Laboratory Animal Center of Ningxia Medical University. Female C57BL/6 mice aged 6–8 weeks were obtained from the Animal Center of Ningxia Medical University. These mice were housed in a specific pathogen-free facility. The animals were accommodated in individual ventilated cages, where the temperature was controlled at 23 ± 1°C and a light cycle of 12 hours of light followed by 12 hours of darkness was maintained. Food and water were provided ad libitum. Throughout the study period, all the animals received compassionate care in accordance with the regulations of the Ministry of Science and Technology of China.

### r*Eg*P29 preparation and animal immunization

The construction and expression of Plasmid *Eg*.P29/pET28a in *Escherichia coli* BL21 (DE3) pLysS, as well as its preservation in our laboratory, have been previously documented [16]. The positive strain was subjected to overnight induction with 1mM isopropyl-β-d-thiogalactoside (IPTG; Invitrogen, Waltham, USA) at 37°C to facilitate the expression of the recombinant protein P29. Subsequently, the purified protein was obtained using a His-bind Purification kit (Merck, Kenilworth, USA) following the manufacturer's instructions, and its purity was assessed by SDS-PAGE (10%). The protein concentration was quantified using a BCA kit (KeyGEN BioTECH, Nanjing, China).

A total of five BALB/C mice were subcutaneously injected with 50 μg of purified r*Eg*P29 protein at three distinct sites on the abdomen. Prior to injection, the r*Eg*P29 protein was emulsified twice with phosphate-buffered saltwater (PBS) and Freund's adjuvant, with a total

volume of 100 μL. The initial immune response was induced using Freund's complete adjuvant, followed by two booster immunizations using Freund's incomplete adjuvant, one week apart. After the last immunization, the mice were allowed to develop an enhanced immune response before they were used for subsequent experiments.

## Generation of anti-r*Eg*P29 monoclonal antibody by hybridoma cells

Two weeks after the administration of booster doses, blood samples were obtained from the tail vein, and serum was assessed for antibody titers using purified r*Eg*P29 through an enzyme-linked immunosorbent assay (ELISA), with preimmune serum serving as a control. Subsequently, the spleen was extracted from mice exhibiting the highest serum titers of anti-P29 antibodies, and splenocytes were fused with mouse myeloma SP2/0 at a 2:1 ratio using electrofusion [43]. Following electrofusion, hybridomas derived from myeloma and mouse spleen cells were resuspended in hypoxanthine-aminopterin-thymidine (HAT) medium and plated in 96-well plates at a density of $2 \times 10^4$ cells/well. Hybridoma cells were cultured in a 96-well microplate at a temperature of 37˚C and a $CO_2$ concentration of 5%. After 10 d, screening tests were conducted to identify positive clones. The supernatants were subjected to ELISA. Hybridomas that tested positive in the ELISA were subcloned using the standard limiting dilution method [25]. After a period–7–10 days, the subcloned cells were observed under a microscope, and wells containing single clonal growth were identified and marked. Single-positive clonal cells were selected for subcloning. Once a 100% positivity rate was achieved, the monoclonal cells were isolated and expanded for culture, leading to the establishment of stable cell lines.

## ELISA assay

ELISA plates were initially coated with r*Eg*P29 at a concentration of 1μg/ml and subsequently incubated overnight at a temperature of 4˚C. The plates were washed five times with PBS-Tween 20 (PBST, 0.05% Tween 20). Subsequently, the plates were blocked for a duration of 1 h at a temperature of 37˚C using 5% skim milk powder in 1×PBST. Serum samples obtained from immunized mice were then added to each well at serial dilutions ranging from 1:1,600 to 1:409,600. The incubation of the samples took place at a temperature of 37˚C for a duration of 1 h. Serum obtained from non-immunized mice was used as a negative control. After five washes with PBST, the plate was covered with a horseradish peroxidase (HRP)-conjugated goat anti-mouse IgG (ab97023, Abcam, Cambridge, MA, USA) diluted 1:10000 in PBST as secondary antibody and incubated for 1h at 37˚C. Then, the plate was washed and incubated with 100 μL of tetramethylbenzidine (TMB, Solarbio Science & Technology Co., Ltd, Beijing,China) per well at room temperature for 8–10 min, and the reaction was terminated using 50 μL of ELISA Stop Solution (Solarbio Science & Technology Co., Ltd, Beijing, China) per well. Finally, the absorbance was measured at 450 nm within 15 min using an ELISA reader (Thermo Fisher Scientific). The dilution of the well with an OD value greater than 2.1 times the OD value of the negative control was determined as the titer of the sample.

The procedure for screening hybridomas for mAbs against P29 was similar to the serum analysis, except that each hybridoma supernatant was added as the primary antibody, and the serum of immunized mice was used as a positive control, while the culture supernatants of myeloma cells (SP2/0) were used as negative controls. An indirect ELISA method was employed to determine the subtypes of the monoclonal antibodies (mAbs) present in the supernatants of the positive hybridomas. The supernatants from positive hybridomas were added as primary antibodies, and HRP-conjugated anti-mouse IgM, IgG1, IgG2a, IgG2b, and IgG3 were added as secondary antibodies.

## Production and purification of antibody

Following ELISA screening, the hybridoma cell line-4G10F4, which exhibited the highest antibody titers, was subjected to extensive serum-free culture for monoclonal antibody (mAb) production. Supernatants from cell cultures were obtained when the cell density reached 90% and were subsequently purified for monoclonal antibodies (mAbs) using a chromatographic column with protein G. The bound antibody was eluted using glycine (0.1 M, pH 2.5). The eluted antibodies were then subjected to overnight dialysis against PBS buffer at a temperature of 4˚C. The concentration of the purified mAbs was determined using the BCA protein assay, and antibody titers were confirmed using ELISA. Additionally, SDS-PAGE was performed to assess the mAb purity.

## Western blot

Total protein was extracted from the hydatid cysts using the Whole Cell Lysis Assay kit (Key-GEN BioTECH, Nanjing, China) according to previously established methods [26]. The extracted samples were subjected to electrophoresis on a 10% SDS-PAGE gel and subsequently transferred onto a 0.2μm PVDF membrane(Millipore, Billerica, MA, USA). To prevent non-specific binding, the membrane was blocked with 5% skim milk for 2 h at room temperature. The primary antibody, r*Eg*.P29 mAb, was then added and incubated overnight at 4˚C. After three washes with PBST, horseradish peroxidase-conjugated goat anti-mouse IgG was added as a secondary antibody and incubated at room temperature for 2 h. Finally, protein expression assays were conducted on a ChemiDoc Touch Imaging System (Bio-Rad Laboratories, Inc., Shanghai, China) using an ECL detection kit (KeyGen Biotech Co., Ltd., Jiangsu, China).

## Antibody epitope identification

The 45 overlapping peptides of r*Eg*P29 were synthesized in sequences consisting of 15 amino acid residues, with an overlap of 10 residues at each end of the polypeptide. These peptides were synthesized and stored in our laboratory [44]. To identify the antigen epitopes of the mAb, indirect ELISA was employed. The peptides were diluted to a concentration of 5μg/mL and used as coating antigens. The prepared purified mAb was used as the primary antibody at a dilution of 1:5,000, and horseradish peroxidase-IgG was used as the secondary antibody.

## Indirect immunofluorescence assay (IFA)

Immunofluorescence staining was performed to determine the localization of P29 in *echinococcus* protoscoleces (PSCs) and the hydatid cyst wall (HCW). The PSCs were washed three times with PBS and fixed with 4% paraformaldehyde(Solarbio Science & Technology Co., Ltd., Beijing,China) for 30 min. Following fixation, the PSCs were washed three times and blocked with 5% sheep serum(ZSGB-Bio, Beijing, China) for 2 h at room temperature. Subsequently, the primary antibody anti-P29 mAb 4G10F4 was added at a dilution of 1:500 and incubated for 1 h at room temperature. The PSCs were then washed three times and incubated with the secondary antibody, Alexa Fluor 647-conjugated goat anti-mouse IgG (1:1000, ab150115, Abcam, Cambridge, MA, USA), at room temperature for 1 h. Following this, the PSCs were washed thrice with PBS and exposed to 4′, 6-diamidino-2-phenylindole (DAPI, Solarbio Science & Technology Co., Ltd, Beijing,China) for a period of 5 to 10 minutes at ambient temperature to facilitate the staining of nuclei. The PSCs were then positioned on a glass slide and gently compressed by placing a cover slip on top and applying slight pressure. Images were acquired using an Zeiss LSM880 confocal microscope (Zeiss, Jena, Germany). HCW detection was performed using the methodology described above.

### SPR assay

Antigen P29 was diluted to six concentrations (500nM, 250nM, 125nM, 62.5, 31.3, and 0nM) in PBST. Monoclonal antibody 4G10F4 was diluted to a concentration of 5 µg/mL using PBST buffer. Subsequently, mAb 4G10F4 was captured using the AMC probe in PBST solution. The resulting solid conjugate was reacted with the antigen P29 and dissociated in PBST buffer. The binding rate, dissociation rate, and affinity constant were determined using a Data Analysis12.0 software (Octet RH16, Sartorius Corporation, Gottingen, Germany).

### Cloning and sequencing of the variable regions of heavy and light chains cDNAs from the selected hybridoma

The total RNA was extracted from 4G10F4 hybridoma cells using TRIzol reagent (Thermo Fisher Scientific, Waltham, MA, USA), followed by cDNA synthesis using a cDNA Synthesis Kit (R312-02, Nuoweizan Biology, Nanjing,China). The cDNA of the heavy and light genes were amplified by PCR using degenerate primers (Zoonbio Biotechnology, Nanjing, China). The resulting amplicons were analyzed by electrophoresis on a 1% agarose gel. The bands of interest were purified and ligated into the pLB vector (VT205; Tiangen,Beijing,China) for cloning. After transformation into competent cells, the positive colonies were screened and subjected to nucleotide sequencing. Finally, the complementarity-determining regions (CDRs) of the mAbs were predicted using the IgBlast tool (https://www.ncbi.nlm.nih.gov/igblast/).

### Isolation of *Ehinococcus multilocularis* protoscoleces

*E. multilocularis* protoscoleces (PSCs) were obtained from the intraperitoneal lesions of BALB/c mice maintained in our laboratory. The metacestode tissues were minced in precooled sterile phosphate-buffered saline (PBS), and the *E.multilocularis* PSCs were filtered using a 180-µm cell strainer into a 50mL sterile centrifuge tube. The initial filtrate was then further filtered using a 100-µm cell strainer, and the resulting filtrate was subsequently filtered with a 40-µm cell strainer to remove calcareous bodies. Finally, the protoscoleces were allowed to settle naturally at the bottom of the container and washed five times with PBS. Viability percentages were determined by calculating the proportion of dead and live protoscoleces within a sample of 300 protoscoleces using an eosin exclusion experiment [21]. Viable protoscoleces were then cultured in Dulbecco's modified Eagle (DMEM) medium (HyClone, Fisher Scientific International Inc.,America) supplemented with 10% heat-inactivated fetal bovine serum (FBS, ExCell Bio, Saiguo Biotech Co., LTD, Guangzhou, China), 1% penicillin-streptomycin, and 10ug of ciprofloxacin/ml(Solarbio Science & Technology Co., Ltd, Beijing, China).

### Isolation of *E. multilocularis* primary cells

In our study, we modified previously established methods [45] for the in vitro cultivation of *E. multilocularis* metacestodes as described [32]. Briefly, metacestodes obtained from the mouse peritoneal cavity were dissected and cut into small tissue blocks measuring approximately 0.5 cm$^3$. The tissue blocks were then washed thrice with sterile PBS. Following this, three of the aforementioned tissue blocks were transferred into a sterile 15mL centrifuge tube, which was filled with 15mL of DMEM medium supplemented with 10% heat-inactivated FBS, 1% penicillin-streptomycin, and 10ug of ciprofloxacin/ml. The tube was securely sealed and incubated at a temperature of 37˚C with a 5% $CO_2$ atmosphere. The medium was replaced every 3–5 days to ensure optimal conditions for cultivation. After a cultivation period of 4–6 weeks, secondary vesicles derived from *E.multilocularis* metacestodes were transferred to 60 mm$^2$ culture dishes

and incubated in a sealed container with an AnaeroPack pouch (Mitsubishi Gas Chemical Co., Inc., Tokyo, Japan). After approximately 2 weeks of anaerobic culturing, intact vesicles measuring 5–10 mm in diameter were collected and dissociated into individual cells using 0.25% trypsin-EDTA (Solarbio Science & Technology Co., Ltd, Beijing, China).

## In vitro assessment of anti-P29 mAb activity against *E. multilocularis* protoscoleces

Viable protoscoleces were transferred to 96-well plates, each well containing 200 PSCs. The PSCs were then incubated with three concentrations (2.5, 5, and 10 μg/mL) of 4G10F4 mAb. Isotype-mouse IgG1 antibody (Bio X Cell, West Lebanon, USA) was used as a negative control at a concentration of 5μg/ml. Albendazole (ABZ, Solarbio Science & Technology Co., Ltd, Beijing, China) was used as a positive control at a concentration of 4 μg/ml, prepared according to the recommended dose [21]. The incubation of protoscoleces took place in an incubator set at 37°C and 5% $CO_2$ for a duration of 24 hours. The viability of PSCs was assessed using a CCK8 kit (Beyotime Biotechnology Co., Ltd, Shanghai, China).

For antibody-dependent cell-mediated cytotoxicity (ADCC) assay, PSCs and three different concentrations of 4G10F4 mAb (2.5, 5, 10 μg/mL) were preincubated in 48-well plates with 400 PSCs per well for 30 min. Peripheral blood mononuclear cells (PBMCs) were freshly isolated from normal mice using Ficoll-Paque PLUS gradient centrifugation(Tianjin HaoYang Biological Manufacture, Tianjin, China) [44]. PBMCs were added at an effector/target ratio of 50:1[46]. In this experiment, PBMCs were used as a negative control. The plates were then incubated at 37°C in a 5% $CO_2$ incubator for a period of 24 hours. PSCs viability was assessed using trypan blue exclusion, and cytotoxicity was determined using an LDH-based cytotoxicity detection kit (Elabscience Biotechnology Co., Ltd., Wuhan, China).

In the complement-dependent cytotoxicity (CDC) assay, PSCs and three different concentrations of 4G10F4 mAb (2.5, 5, 10 μg/mL) were mixed respectively in 96-well plates. Baby rabbit complement, obtained from Cedarlane (Burlington, Canada), was utilized as the complement source [47] and added at a dilution ratio of 1:50 [46]. The plates were incubated at 37°C in a 5% $CO_2$ incubator for a duration of 24 hours. The viability of the PSCs was subsequently assessed using a CCK8 kit.

## Assessment of anti-P29 mAb in vitro toxicity in *E. multilocularis* primary cells

*E. multilocularis* primary cells were obtained from metacestode vesicles cultured in vitro, as described above. Cells were plated into 96-well plates with $1\times10^5$ per well and incubated with 4G10F4 mAb at concentration of 5 μg/ml. For the CDC assay, cells and 4G10F4 mAb (5 μg/mL) were mixed in 96-well plates with rabbit complement at a dilution ratio of 1:50. Isotype-mouse IgG1 antibody (5 μg/ml) was used as a negative control and ABZ (4 μg/mL) was used as a positive control. The plate was incubated in a sealed container with an AnaeroPack pouch at 37°C and 5% $CO_2$ for 24 hours. Cell viability was determined by a CCK8 kit.

Live/dead cells were detected using 5-(6)-carboxy-fluorescein succinimidyl ester (CFSE) and propidium iodide (PI) staining. Primary cells of *E. multilocularis* were subjected to labeling with 1 μM CFSE (BD Bioscience, Franklin Lakes, NJ, USA) for a duration of 10 minutes at a temperature of 37°C in phosphate-buffered saline (PBS). The labeling process was terminated by adding a 10-fold volume of PBS and subsequently washing twice with PBS before seeding into 24-well plates at a density of $1\times10^6$ cells per well. CFSE-labeled cells were subsequently incubated with 4G10F4 mAb (5 μg/mL) and rabbit complement at a dilution ratio of 1:50 for 24 h in a hermetically sealed container containing an AnaeroPack pouch. PI (Nanjing

Jiancheng Bioengineering Institute, Nanjing, China) was used to determine the proportion of cells that died. The samples were then examined by fluorescence microscopy (EVOS FL; Thermo Fisher Scientific, Waltham, Massachusetts, USA) and analyzed using ImageJ software. The percentage of cytotoxicity was determined using the following formula: % cytotoxicity = (number of propidium iodide-labeled CFSE-expressing cells/total number of CFSE-expressing cells)×100 [48].

## Determination of the in vivo effects of anti-P29 mAb on *E. multilocularis* metacestode

Two experiments were conducted to assess the in vivo therapeutic effectiveness of anti-p29 mAb against *E. multilocularis* metacestodes. In the first experiment, mice were intraperitoneally injected with *E. multilocularis* protoscoleces and subsequently administered three different doses of anti-P29 mAb once per week via tail vein injection at 12 weeks postinfection. The objective of this study was to evaluate the therapeutic potential of the anti-p29 mAb and to identify the optimal dosage for combating *E. multilocularis*. In the second experiment, mice were infected with *E. multilocularis* protoscoleces via portal vein injection. One week post-infection, the mice were treated with the optimal therapeutic dose of anti-P29 mAb, as determined in the first experiment, to confirm its therapeutic efficacy in vivo.

In experiment 1 (Fig 5A), C57BL/6 mice were intraperitoneally injected with 2000 PSCs in 200 μL 0.9% sodium chloride (NS), while the blank group received an equivalent volume of saline. After 12 weeks, mice with metacestodes measuring 3-5mm$^3$ (as determined by abdominal ultrasonography) were randomly assigned to five groups, each consisting of nine mice. These groups included the isotype-control group, which was treated with isotype-mouse IgG1 antibody (5 mg/kg); the ABZ group, which received oral administration of 50 mg/kg ABZ dissolved in 0.2 mL) once daily; and the three experimental groups, namely the anti-P29 mAb 2.5/5/10 groups, which were subjected to treatment with anti-P29 mAb at varying dosages of 2.5, 5, and 10 mg/kg, dissolved in 0.2mL of normal saline, administered once per week through tail vein injection. Normal mice were injected with normal saline weekly as a sham group. After 8 weeks of treatment, the mice were euthanized by cervical dislocation under isoflurane anesthesia. Blood samples were collected from the eyeballs before euthanasia and centrifuged at 3500 r.p.m for 10 min at 4°C. The serum was harvested and stored at -20°C for inflammation cytokines detection by using Cytometric Bead Array(CBA) mouse inflammation kit (552364,Becton,Dickinson and Company, New Jersey, USA) and liver and kidney function tests. Metacestode tissues, spleen, and liver were carefully harvested and weighed. The metacestode inhibition rate (%) was calculated as follows: (total wet weight of metacestode tissue in the control group–total wet weight of metacestode tissue in the treatment group) / total wet weight of metacestode tissue in the control group×100. Liver and spleen indices were calculated using the following formulas: liver index = liver weight/body weight, and spleen index = spleen weight/body weight.

In experiment 2 (Fig 6), C57BL/6 mice were anesthetized by intraperitoneal injection of tri-bromoethanol (Lab Animal Technology Develop Co., Beijing, China) and inoculated through the hepatic portal vein as described [37] with 1000 PSCs in 100 μL NS, while the sham group received an equivalent volume of saline. One week after surgery, the mice were randomly assigned to two groups (n = 5): the isotype-control group, which was treated with isotype-mouse IgG1 antibody, and the experimental group, which received 5 mg/kg anti-P29 mAb once per week via tail vein injection. After 8 weeks of treatment, serum was harvested for the detection of inflammatory cytokines. The spleens and livers were carefully harvested and weighed. The liver surfaces were carefully screened, and hepatic lesions were counted to assess

the intensity of *E. multilocularis* infection in different groups. Part of the liver containing the infected lesions was fixed in 4% paraformaldehyde for pathological examination.

## Statistics

The data were analyzed and processed using GraphPad Prism software 8.0 (GraphPad Software), and presented as mean ± SD. Student's t-test was used to analyze data from two groups, while one-way ANOVA was used to analyze data from multiple groups. Statistical significance was set at $P < 0.05$.

## Supporting information

**S1 Data. Excel spreadsheet with numerical data for Figs 1B, 1C, 2D, 3B, 3C, 3D, 4B, 4D, 5C, 5D, 5E, 7C, 7D and 7E.**
(XLSX)

**S1 Fig. Purified recombinant P29 protein was analyzed by SDS-PAGE.** Lane M: protein marker; Lane 1: *Escherichia coli* containing pET28a-P29 before IPTG induction; Lane 2: *E. coli* containing pET28a-P29 6h after IPTG induction; Lane 3: purified rEg.P29 using His-affinity chromatography.
(TIF)

**S2 Fig. Diagram of interaction between 4G10F4 mAb and P29.** The antibody (blue protein) and P29 (yellow protein) form hydrogen bonds through amino acid residues such as TYR-103, HIS-105 and ASN-75 (yellow dashed line).
(TIF)

**S3 Fig. The affinity of 4G10F4 antibody for P29 were determined by surface plasmon resonance (SPR).** Antigen P29 was diluted to six concentrations (500 nM, 250 nM, 125 nM, 62.5 nM, 31.3 nM, and 0 nM).
(TIF)

**S4 Fig. The in vitro effects of 4G10F4 on *E. multilocularis* protoscoleces.** (A, B, C) The morphological characteristics of protoscoleces were examined using an optical microscope 48 hours post-intervention. Upon visual inspection, no discernible variations in protoscoleces were observed between the different groups. (D) Comparion of LDH activity in protoscoleces culture supernatant of different treatment groups. ns: no statistical significance.
(TIF)

**S5 Fig. Representative murine abdominal B-ultrasound images at 8 weeks post infection.** The abdominal ultrasound images revealed the presence of multilocular cystic masses, identified as metacestode lesions, characterized by the presence of echogenic thin septa, as denoted by the red arrows.
(TIF)

**S6 Fig. The levels of serum inflammatory cytokines.** (A-F) IL-6, IL-10, MCP-1, IFN-γ, TNF, and IL-12p70, were compared between groups in an intraperitoneal infected mice model following treatment.
(TIF)

**S7 Fig. Safety evaluation of 4G10F4 therapy.** (A,B) Comparison of alanine aminotransferase (ALT) and aspartate aminotransferase (AST) serum concentrations between groups after treatment. (C and D) The concentrations of blood urea nitrogen (BUN) and creatinine (CREA) were analyzed between groups following treatment in an intraperitoneal infected mice model.

(E) The body weight was monitored throughout the course of the experiment involving the treatment of intraperitoneal infected mice. (F) The body weight was monitored during the experiment involving the treatment of mice infected through the portal vein.
(TIF)

**S1 Table. ELISA results of mouse antiserum to r*Eg*P29.**
(XLSX)

**S2 Table. Fusion cell screening results.**
(XLSX)

**S3 Table. Subclonal cells screening results.**
(XLSX)

**S4 Table. Subclonal cells secondary screening results.**
(XLSX)

**S1 File. The sequencing results light chain (L) from 4G10F4 mAb.** (Open with SnapGene software).
(ZIP)

**S2 File. The sequencing results of heavy chain (H) from 4G10F4 mAb.** (Open with Snap-Gene software).
(ZIP)

## Acknowledgments

We thank the classmates and teachers in the laboratory for the help during our experiments.

## Author Contributions

**Conceptualization:** Zihua Li, Wei Zhao.

**Formal analysis:** Cuiying Zhang, Siyu Hou, Zihua Li.

**Funding acquisition:** Tao Li, Zihua Li, Wei Zhao.

**Investigation:** Cuiying Zhang, Tao Li, Siyu Hou, Jing Tang, Rou Wen, Chan Wang, Shiqin Yuan, Zihua Li.

**Methodology:** Cuiying Zhang, Tao Li, Shiqin Yuan, Zihua Li.

**Resources:** Wei Zhao.

**Software:** Cuiying Zhang, Siyu Hou.

**Supervision:** Zihua Li, Wei Zhao.

**Validation:** Jing Tang, Shiqin Yuan.

**Writing – original draft:** Cuiying Zhang, Zihua Li.

**Writing – review & editing:** Cuiying Zhang, Zihua Li, Wei Zhao.

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
