## [Decision Letter · Decision Letter 0]

5 Jun 2024

Dear Mr Zhao,

Thank you very much for submitting your manuscript "Enhancing the Therapeutic Potential of P29 Protein-Targeted Monoclonal Antibodies in the Management of alveolar echinococcosis through CDC-Mediated Mechanisms" for consideration at PLOS Pathogens. As with all papers reviewed by the journal, your manuscript was reviewed by members of the editorial board and by several independent reviewers. In light of the reviews (below this email), we would like to invite the resubmission of a significantly-revised version that takes into account the reviewers' comments.

We cannot make any decision about publication until we have seen the revised manuscript and your response to the reviewers' comments. Your revised manuscript is also likely to be sent to reviewers for further evaluation.

Sincerely,

Edward Mitre

Academic Editor

PLOS Pathogens

Meera Nair

Section Editor

PLOS Pathogens

Michael Malim

Editor-in-Chief

PLOS Pathogens

orcid.org/0000-0002-7699-2064

Reviewer's Responses to Questions

**Part I - Summary**

Reviewer #1: This study presents a significant advancement in the treatment of alveolar echinococcosis (AE), a lethal helminth infection, by developing monoclonal antibodies targeting the P29 protein of Echinococcus multilocularis. The study highlights the exceptional potential of one such antibody, mAb 4G10F4, in inhibiting the viability of E. multilocularis protoscoleces and primary cells in vitro and reducing parasite burden in vivo in mouse models. This inhibition is primarily attributed to the antibody-mediated complement-dependent cytotoxicity (CDC) mechanism. The research validates 4G10F4's safety and efficacy, suggesting it as a promising therapeutic option against AE, comparing favorably to traditional treatments with benzimidazoles which have limited efficacy and adverse effects. However, there are some concerns should be addressed before publication.

Reviewer #2: Overall this is an interesting manuscript providing convincing evidence that a monoclonal antibody directed against an Echinococcus antigen has anti-parasite activity against metacestode stages of the parasite. The in vivo models of E. multilocularis infection and mAb activity are a strength of the study.

Reviewer #3: The manuscript by Zhang et al. describes the identification of a P29-targeting monoclonal antibody, which was assessed for its efficacy against E. multilocularis in vitro and in vivo. The manuscript is well written and provides novel, interesting data. However, there are several points that should be addressed by the authors. My major critique concerns the two in vivo experiments. Based on the results following hepatic portal vein infection with protoscolices it seems that the mAb treatment one week after infection prevents the establishment of the infection, while its efficacy against the developed metacestode stage (12 weeks after i.p. infection) is less prominent. Given that two different methods of infection were used, this has to be investigated in further detail.

**Part II – Major Issues: Key Experiments Required for Acceptance**

Reviewer #1: 1. In figure 2D, what does the highest peak next to the Blank group represent?

2. Please provide deeper insights into the mechanism of action of 4G10F4, especially its interaction with the immune system beyond CDC.

3. What are the long-term effects and safety of 4G10F4 treatment? If the authors have done any extend research for evaluation?

Reviewer #2: 1. My main concerns with the experimental data center around Figure 4. Specifically, where in the parasite do these primary cells originate from, and can you confirm these cells actually express P29? P29 expression within the parasite appears to be tissue-specific, as it is absent from cyst fluid, so it is possible that the primary cells do not express it, or do not express it uniformly. An immunoblot of primary cell lysates probed with the 4G10F4 mAb would partly address this concern. It would also be helpful to know what percentage of cells express P29, and where in the cells the P29 antigen is located - e.g. is it membrane-associated? Is it expressed on the cell surface? Or in intracellular vesicles? Perhaps some immunofluorescence microscopy would be appropriate to address these concerns.

2. The in vivo experiments are a strength of the manuscript, but it is important to show these results are reproducible. At a minimum, the authors should show that the results have been recapitulated in two independent experiments.

3.An ethics statement, indicating the animal experimentation was reviewed and approved by an appropriate institutional review committee before the work was undertaken, is required.

Reviewer #3: - The impact of mAb treatment on the development of the metacestode stage development should be also assessed when mice are treated 1 week after intraperitoneal infection. Similarly, given that metacestode tissue can be easily cultured and produces vesicles in vitro, it should be tested whether the mAb (+/- complement) prevents the release of vesicles in vitro. Optimally, you could infect mice/gerbils with these in vitro treated metacestodes to confirm that the treatment prevents the establishment of new infections / spreading of the metacestode stage.

- Fig. 2: Given that you are focusing in this manuscript on E. multilocularis, the binding of the mAb to vesicles and metacestode tissues of E. multilocularis should be shown as well.

- Does the mAb (+/- PBMCs, complement) impact the protruding of the protoscolices?

**Part III – Minor Issues: Editorial and Data Presentation Modifications**

Reviewer #1: 1. Figure 2E, please label clearly the species of P29 and clarify how the putative three-dimensional structures of P29 and the anti-P29 antibody were obtained. Please also label the CDRs on the antibody structure. If they are also close to the epitope on P29? Also, please provide a complete protein sequence for the antibody in addition to the gene sequencing results provided, which will be helpful to better understand the protein feature instead of searching multiple sequencing results.

2. SPR binding curve should be remade by using the exported raw data instead of a screenshot.

Reviewer #2: Line 157 and Fig 1C - what is the relevance of titers here? In what source material? Hybridoma supernatant? If so, the "titer" will depend on variables related to culture of the hybridoma cells, which is not really relevant. Unless you normalized the amount of each antibody in this analysis, I am not certain the data add anything to the manuscript.

Fig.1D - not sure the results from this experiment can be interpreted as quantitative - I would avoid that language at line 159 etc.

Fig. 2B, 2C - is the parasite material E. granulosus or E. multilocularis? Please clarify

Line 186 - "SPR" - please define (presumably surface plasmon resonance, but please clarify to avoid ambiguity)

Fig 3 - why use three different methods to assess protoscolex viability? Consistency in methodology would have allowed a direct comparison between these data. As it stands, I'm not sure how they compare. Was an isotype control antibody with irrelevant specificity used as the negative control?

Fig. 5 - sham data for C and D would be helpful to see. Fig. 5G-5L - what is the "blank" group?

Line 329 - "CE2"?

Discussion - first part is a reiteration of points made in the Introduction. Discussion could be made more concise elsewhere too, e.g. there is no need to provide a detailed discussion of antibody structure and function, as this is well established.

Reviewer #3: - The different impact on the acute versus established infection should be clearly highlighted in the abstract and the discussion.

- Line 423f: Please show the data that demonstrates that mAb and complement treatment of primary E. multilocularis cells prevented the formation of clones.

- Please discuss your reasoning of the selected treatment regimen (once per week).

- Please discuss and reference the potential use of IgG1 versus IgG2 monoclonal antibodies for therapeutic approaches.

- The description of the in vitro and in vivo results should be rephrased in the abstract. Clearly indicate that CDC is required for mAb efficacy in vitro and highlight the differences in both in vivo experiments.

- As the methods section is at the end of the article, it is necessary to shortly introduce some methods/readouts already in the results section (as mentioned below).

- Please check the English by a native speaker.

- Line 96: “Clostridium difficile” in italics

- Fig. 1 B, C: Please use colors to facilitate the differentiation of the different IgG classes / clones.

- Fig. 2B: Line 164 to 165. Pease delete the sentence “The purified mAb exhibited …” and only state that the mAb binds to PSC and HCW, but not HCF, as you did in the following sentence.

- Fig. 4: Please explain the origin of E. multilocularis primary cells already in the results section. Please increase the size of Fig. 4A, as it is not possible to detect the described differences.

- Line 235: please indicate the treatment regimen used.

- Figure 6 and 8 legend: indicate the number of mice used per group and number of repeat experiments and whether the mean or median is shown.

- Line 243: Please delete the sentence “A similar effect was seen…”. This is not supported by the shown data.

- Line 247f.: Please explain shortly in the results section how you calculate the spleen and liver index.

- Line 249ff: How did you assess the metacestode inhibition rates and where is this data shown? Do you refer to Fig. 6B? Please clarify.

- I recommend to combine Fig. 5 and Fig. 6A-D into one figure and Fig. 6E-L as additional figure. This way the figures fit to the paragraphs in the results section.

- Fig. 6G-L: Similar to the ALT and AST values, cytokine levels should be compared to sham mice, not blank controls. The x-axis should not show negative cytokine levels.

- Fig. 8A, B, F: Increase the size of these figures, to facilitate the identification of the metacestodes. E.g., it is currently not possible to identify the PAS-positive sections.

- I recommend to combine Fig. 7 and 8.

- Please rephrase the term “secondary infection”. Clearly state the experimental design (e.g. i.p. infection with protoscolices followed by i.v. mAb treatment)

- Line 327: add reference

- Line 397-410: I recommend to mention this shortly in the results section (instead of the discussion) to explain the selection of the used methods.

PLOS authors have the option to publish the peer review history of their article (what does this mean?). If published, this will include your full peer review and any attached files.

Reviewer #1: No

Reviewer #2: No

Reviewer #3: No
---

## [Editor Report · Decision Letter 1]

5 Aug 2024

Dear Mr Zhao,

We are pleased to inform you that your manuscript 'Enhancing the Therapeutic Potential of P29 Protein-Targeted Monoclonal Antibodies in the Management of alveolar echinococcosis through CDC-Mediated Mechanisms' has been provisionally accepted for publication in PLOS Pathogens.

Best regards,

Edward Mitre

Academic Editor

PLOS Pathogens

James Collins III

Section Editor

PLOS Pathogens

Michael Malim

Editor-in-Chief

PLOS Pathogens

orcid.org/0000-0002-7699-2064
---

## [Editor Report · Acceptance letter]

20 Aug 2024

Dear Mr Zhao,

We are delighted to inform you that your manuscript, "Enhancing the Therapeutic Potential of P29 Protein-Targeted Monoclonal Antibodies in the Management of alveolar echinococcosis through CDC-Mediated Mechanisms," has been formally accepted for publication in PLOS Pathogens.

Best regards,

Michael Malim

Editor-in-Chief

PLOS Pathogens

orcid.org/0000-0002-7699-2064